# Simulating the spatiotemporal variations in aboveground biomass in Inner Mongolian grasslands under environmental changes

Guocheng Wang[1], Zhongkui Luo[2], Yao Huang[3], Wenjuan Sun[3], Yurong Wei[4], Xi Deng[5], Jinhuan Zhu[6], Tingting Li[1], Wen Zhang[1]

[1]LAPC, Institute of Atmospheric Physics, Chinese Academy of Sciences, Beijing, 100029, China.
[2]College of Environmental and Resource Sciences, Zhejiang University, Hangzhou 310058, Zhejiang, China.
[3]State Key Laboratory of Vegetation and Environmental Change, Institute of Botany, Chinese Academy of Sciences, Beijing 100093, China.
[4]Inner Mongolia Ecology and Agrometeorology Centre, Hohhot, Inner Mongolia 100051, China.
[5]School of Atmospheric Sciences and Guangdong Province Key Laboratory for Climate Change and Natural Disaster Studies, Sun Yat-sen University, Zhuhai 519000, China
[6]LAOR, Institute of Atmospheric Physics, Chinese Academy of Sciences, Beijing, 100029, China.

*Correspondence to*: Guocheng Wang (wanggc@mail.iap.ac.cn)

**Abstract.** Grassland aboveground biomass (AGB) is a critical component of the global carbon cycle and reflects ecosystem productivity. Although it is widely acknowledged that dynamics of grassland biomass is significantly regulated by climate change, *in situ* evidence at meaningfully large spatiotemporal scales is limited. Here, we combine biomass measurements from six long-term (> 30 years) experiments and data in existing literatures to explore the spatiotemporal changes in AGB in Inner Mongolian temperate grasslands. We show that, on average, annual AGB over the past four decades is 2,561 kg ha$^{-1}$, 1,496 kg ha$^{-1}$ and 835 kg ha$^{-1}$, respectively, in meadow steppe, typical steppe and desert steppe in Inner Mongolia. The spatiotemporal changes of AGB are regulated by interactions of climatic attributes, edaphic properties, grassland type and livestock. Using a machine learning-based approach, we map annual AGB (from 1981 to 2100) across the Inner Mongolian grasslands at the spatial resolution of 1 km. We find that on the regional scale, meadow steppe has the highest annual AGB, followed by typical and desert steppe. The future climate change characterized mainly by warming could lead to a general decrease in grassland AGB. From the perspective of climate change, on average, compared with the historical AGB (i.e., average of 1981-2019), the AGB at the end of this century (i.e., average of 2080-2100) would decrease by 14% under RCP4.5 and 28% under RCP8.5, respectively. If the carbon dioxide ($CO_2$) enrichment effect on AGB is considered, however, the estimated decreases in future AGB can be reversed due to the growing atmospheric $CO_2$ concentrations under both RCP4.5 and RCP8.5. The projected changes in AGB show large spatial and temporal disparities across different grassland types and RCP scenarios. Our study demonstrates the accuracy of predictions in AGB using a modelling approach driven by several readily obtainable environmental variables; and provides new data at large scale and fine resolution extrapolated from field measurements.

## 1 Introduction

Grassland occupies ~40% of the world land and is an essential component of global terrestrial ecosystems (Hufkens et al., 2016). Grassland provides plenty of ecosystem services such as suppling food to livestock therefore meat and milk to humans

(Sattari et al., 2016) and accumulating carbon from atmosphere thus mitigating global warming (O'Mara, 2012). All of these functions are more or less directly dependent on grassland biomass, which has been recognized significantly influenced by environmental changes and anthropogenic activities (Hovenden et al., 2019). Thus, quantifying the dynamics of grassland biomass and revealing the underlying mechanisms are of fundamental importance (Andresen et al., 2018).

Dynamics of grassland aboveground biomass (AGB) are driven by complex interactions among a series of environmental attributes such as climate variables (De Boeck et al., 2008;Wang et al., 2020a). The magnitudes and directions of climate change effects on AGB can vary across different local environments as well. For example, climate warming can either avail to AGB accumulation through reducing constraints of low temperature (Gonsamo et al., 2018;Park et al., 2019) or go against AGB formation by aggravating water stress on plant growth (Fan et al., 2009;Hu et al., 2007). In addition, in most existing studies, the mean annual climate attributes (e.g., temperature and precipitation) have widely been treated as potential drivers on spatiotemporal variations of grassland biomass (Fan et al., 2009;Ma et al., 2008). However, growing evidences have demonstrated the importance of seasonality and intra-annual variability of climate in regulating the biomass dynamics (Godde et al., 2020;Grant et al., 2014). For example, Peng et al. (2013) reported that variations in seasonal precipitation significantly alter the annual NPP in Inner Mongolian grasslands. To date, climatic seasonality and intra-annual variability have seldom been considered in assessing grassland AGB, particularly at large extents of space and time. Moreover, recent studies have suggested the possible co-regulating effects of soil properties (Bhandari and Zhang, 2019;Jia et al., 2011), grassland type and grazing intensity (Eldridge and Delgado-Baquerizo, 2017) on AGB, which have also seldom been included in exploring the spatiotemporal changes in grassland AGB. Comprehensively considering these covariates, rather than including only a few mean annual climatic attributes, provides an opportunity to more accurately predict grassland AGB dynamics and disentangle the response of AGB to the complex interactions between environmental drivers.

Inner Mongolian grasslands account for more than half of China's northern temperate grassland area (Department of Animal Husbandry and Veterinary, 1996) and have the nation's largest grassland biomass carbon stock (Piao et al., 2004). The annual productivity in this region tends to vary in response to climate change (Bai et al., 2008). Since the start of 1980s, warming has been taking place in many parts of Inner Mongolia (Wang et al., 2019). Under this temperature rising, the spatiotemporal variations in grassland AGB, however, is still unclear. Although efforts have been taken to quantify AGB dynamics at the regional scale, these studies used mainly remote-sensing approaches and generally showed large disparities (Guo et al., 2016;Long et al., 2010;Ma et al., 2010a). Evidence from datasets independent of remote-sensing products can certainly contribute to the assessments of spatiotemporal dynamics of AGB at the regional scale. In addition, the climate in the future is projected to experience substantial changes (IPCC, 2007) and thus significantly affect grassland AGB dynamics while little is known about the fate of AGB under future climate changes. Furthermore, it has been reported that carbon dioxide ($CO_2$) enrichment may increase plant productivity through enhancing photosynthetic rates and reducing stomatal conductance thereby increasing water use efficiency (Fay et al., 2012;Pastore et al., 2019). This might provide an opportunity to mitigate or even

reverse the harmful effects of other environmental changes on grassland AGB (Lee et al., 2010), e.g., the enhanced water limitations resulted from climate warming. The actual effects of $CO_2$ enrichment on AGB, however, depend substantially on local environmental factors such as water availability (Brookshire and Weaver, 2015) and soil texture (Polley et al., 2019).

In this study, we collate a comprehensive dataset of *in situ* measurements on plant biomass and climatic records in Inner Mongolian grasslands from six long-term experiments and those data from existing literatures. We calibrate and validate a machine learning-based model for predicting the aboveground biomass in the study region, by treating tens of environmental covariates (climates, soils, livestock, and grassland type) as predicting variables. Then, we map the annual aboveground biomass at a spatial resolution of 1 km in over the periods of 1981-2019 (using historical climatic dataset) and 2020-2100 [using climate projections driven by two representative concentration pathways (e.g., RCP4.5 and RCP8.5)]. We also include the possible effects of atmospheric $CO_2$ enrichment on future grassland AGB dynamics in the study region.

## 2 Materials and Methods

### 2.1 Study region and datasets of grassland aboveground biomass

The study region (i.e., Inner Mongolian grasslands) is characterized mainly by a temperate climate (Zhang et al., 2020b) and thus is named Inner Mongolian temperate grasslands as well, which can be generally classified into three categories, i.e., meadow steppe, typical steppe and desert steppe (National Research Council, 1992). In brief, meadow steppe distributes mainly in the eastern areas, typical steppe locates mostly in the central Inner Mongolia, and desert steppe is found mainly to the west of typical steppe (Fig. 1). In this study, we acquired two datasets of *in situ* aboveground biomass (AGB) in Inner Mongolian grasslands. First, we obtained the AGB at six long-term (i.e., more than 30 years) experimental sites across the study region (Fig. 1, Data S1). These six sites were established by Inner Mongolia Meteorological Bureau of China at early 1980s, measurements of AGB at each site has been carried out year by year since then. At each site, four fenced plots (i.e., four replicates) were set up to collect plant biomass data during plant growing seasons (e.g., from May to September). For each measurement replicate, the plants within a one square meter area were clipped and collected in a cloth bag. The samples were further air-dried to constant weights (weighted once every three days until the percent change in two consecutive weights are less than 2%). It is noted that plant growth rate could peak at different periods across time and space. Following Scurlock et al. (2002), we determined the annual plant biomass as the largest observed monthly biomass during a year (normally at the end of August at Ergun and at the end of September at other three sites). Apart from measurements of these six long-term field experiments, we also retrieved a dataset of grassland AGB from Xu et al. (2018), who recently conducted a thorough literature synthesis and established a comprehensive dataset of plant biomass in the grasslands of northern China. For the dataset constructed by Xu et al. (2018), we used only the observations conducted in Inner Mongolian grasslands and with investigation time and coordinates clearly reported (Fig. 1). In general, the grassland AGB derived from these two different datasets (i.e.,

long-term experiments and literature synthesis) are comparable (Fig. S1). In total, we obtained 511 individual measurements across 247 locations in the study region (Fig. 1, Data S1).

## 2.2 Environmental covariates

Environmental covariates including climate, soil, grassland type and livestock were retrieved for both AGB driver assessment and machine learning-based model fitting. For climatic covariates, we first obtained the daily climatic records of 120 climatic stations established in Inner Mongolia (Fig. 1) from the National Meteorological Information Centre (NMIC) of China. The daily climatic attributes such as minimum, average, and maximum temperature and precipitation were transformed into monthly time series data using the *daily2monthly* function in the R package *hydroTSM*. Based on these monthly data, we calculated 23 bioclimatic variables (Table 1) with an annual time step over the period of 1981-2019 by using the *biovars* function in the R package *dismo*. By doing so, we aim to comprehensively consider the possible effects of seasonality, intra- and inter-annual variability of climates (Fick and Hijmans, 2017) on grassland AGB. By further applying an interpolation algorithm (Thornton et al., 1997) to these 23 bioclimatic variables at the 120 stations, we created the raster layers of the climatic attributes with a spatial resolution of 1 km year by year. For the edaphic covariates, we directly extracted 10 raster soil layers representing key soil physical and chemical properties (Table 1) at a 1 km spatial resolution in the study region from ISRIC-WISE soil profile database (Batjes, 2016).

The grazing intensity in this study was represented by the quantity of three key livestock (i.e., cattle, goat and sheep; Table 1) because they are the majority in Inner Mongolian grasslands (National Bureau of Statistics of China, 1981-2019). Here, we first derived the regional distribution data for cattle (Fig. S2 a), goat (Fig. S2 b) and sheep (Fig. S2 c) during 2010 in the study region from Gilbert et al. (2018). Then, we obtained the yearly total amount of each livestock in the study region (Fig. S2 d) from National Bureau of Statistics of China (1981-2019). By assuming a similar spatial distribution of livestock over time, we generated raster layers of each of the three livestock year by year over the past four decades using the above-mentioned two datasets. In addition, a spatial layer of grassland type (i.e., meadow steppe, typical steppe and desert steppe; Fig. 1 and Table 1) at 1 km resolution was derived from the Vegetation Map of China (Zhang, 2007), the digital version of which is publicly obtainable (http://data.casearth.cn/sdo/detail/5c19a5680600cf2a3c557b6b).

## 2.3 Machine learning models to predict grassland AGB

To predict grassland aboveground biomass (AGB) across the region, we generated a suite of machine learning-based predictive models for AGB treating edaphic and climatic variables, grassland type and livestock (Table 1) as candidate predictors. Here, data from the 511 measurements (Fig. 1 and Data S1) were used to fit the models. For the spatial layers of soil properties and grassland type, which were assumed to be constant over time, we retrieved the associated covariates using the geographical coordinates of the 511 measurements. For those variables varying over time (e.g., climatic variables and livestock), we extracted the associated attributes using both the locations and investigating year of the 511 measurements. In fitting the models,

AGB is treated as a dependent variable and the environmental covariates (Table 1) are treated as independent variables. Before fitting the models, we converted the categorical variables (i.e., grassland type) to dummy variables. This is to avoid simply deducing the dependent variables in a certain category using the independent variables (e.g., climate variables) in other categories in building the models. Then, the function *findCorrelation* in R package *caret* was used to exclude the environmental covariates with high multicollinearities. Following Brownlee (2016), the remaining attributes were further adopted in model training (80% stratified samples) and validation (the remaining 20% stratified samples). We used a 10-fold cross-validation to train a suite of machine learning models using three algorithms [i.e., random forest (RF), Cubist and support vector machines (SVM)], which are implemented in the R package *caret*. The amount of variance in AGB explained by each model was assessed by the coefficient of determination ($R^2$). The root mean square error (RMSE, kg ha$^{-1}$) was also calculated (RMSE = $\sqrt{\sum_{i=1}^{n} \frac{(P_i - O_i)^2}{n}}$, where $n$ is sample size, $P_i$ and $O_i$ are the $i_{th}$ predicted and observed AGB, respectively) to compare the model simulations and observations. Apart from the three individual machine learning-based models, we also derived an ensemble model by adopting a principal component analysis (PCA) approach based on the predictions of the above-mentioned three models. In brief, the smaller an individual model's RMSE is, the more the model's outputs contribute to the ensemble predictions.

## 2.4 Assessment of drivers on AGB

We used three approaches to explore the effects of environmental covariates on grassland AGB. First, the machine learning models themselves provide assessments of the relative importance (RI) of each independent variable in predicting the dependent variable (e.g., grassland AGB in this study). In general, the greater the RI of a variable is, the larger its influence on AGB is. Second, we adopted the Mantel test (Mantel, 1967) to assess the relationship between similarity of different grassland types and the similarity of environmental covariates using the R package *vegan*. Here, the standardized Mantel's r (ranges from 0 to 1) is used to represent the strength of this relationship (the higher the Mantel's r is, the stronger the correlation is) and the associated significance is indicated by the *P* value determined from 999 randomization (Legendre and Fortin, 1989). Third, we conducted a path analysis by using three latent variables, i.e., climate, soil and livestock, to evaluate their regulating effects on AGB. For each latent variable of climate and soil, the specific indicators were pre-identified using the above-mentioned R function *findCorrelation* to exclude those attributes with high multicollinearities. In constructing the inner model matrix of the path model, we hypothesized all the three latent variables have direct effects on AGB and climate may also indirectly affect the dependent variable through influencing soil properties (Luo et al., 2019). Here, we adopted the partial least squares (PLS) approach (Sanchez, 2013) and used the R package *plspm* to perform the path analysis. In interpreting the path analysis results, it is noted that the loadings of an indicator show the correlations between a latent variable and its indicators. All the indicators were standardized before the path analysis was performed.

## 2.5 Regional mapping and uncertainty analysis

Using the fitted machine learning-based ensemble model, we mapped AGB in Inner Mongolian grasslands (at a spatial resolution of 1 km) on an annual time step in the history (1981-2019) and future (2020-2100). In mapping the historical AGB, the model is run using environmental covariates extracted from the regional data layers (see *Environmental covariates*). Prediction uncertainty was quantified using a Monte Carlo analysis to develop the probability density functions (PDF) for each edaphic, climatic and livestock variable within the ranges of mean ±10%. The ensemble machine learning model was then run for 200 times in each grid with each of independent variables assigned from the PDF. The average and coefficient variation (CV, calculated as the standard deviation divided by the average) were then determined in each grid using the 200 model outputs to represent the predicted AGB and the associated uncertainty, respectively.

For predictions of AGB in the future (i.e., 2020-2100), we included the climatic datasets projected by a typical CMIP5 global circulation model, i.e., CESM1-BGC, which was run by National Center for Atmospheric Research (NCAR). Here, we directly obtained the processed climatic products constructed by Karger et al. (2020), who recently generated a downscaled and bias-corrected temperature and precipitation datasets. Specifically, these future climatic datasets were driven by two scenarios of representative concentration pathways (RCP4.5 and RCP8.5) at monthly step in this century. According to the model projections, mean annual temperature (MAT) under both RCPs will continue to increase in the following decades (Fig. S3 a). The extent of climate warming is generally higher under RCP8.5 than that under RCP4.5 (Fig. S3 a). The mean annual precipitation under both RCPs show large inter-annual variabilities (Fig. S3 b). After obtaining the future climate datasets, we also use the *biovars* function in R environment (see *Environmental covariates*) to calculate the 23 interested bioclimatic attributes (Table 1) for both RCPs year by year from 2020 to 2100. In projecting the future AGB dynamics using the ensemble machine learning model, we assume that the soil properties will not significantly change over time and current grazing intensity will keep relatively stable (i.e., the average amount of livestock during 2014-2019 is used in future predictions). In addition, the uncertainty analysis for future AGB predictions were performed using the same approach as that adopted in mapping the historical AGB. Moreover, the $CO_2$ concentrations have been projected to increase under the two RCPs (i.e., RCP4.5 and RCP8.5) used in this study (Fig. S4 a). The growing $CO_2$ concentrations can either increase AGB through enhanced photosynthetic rates (Fay et al., 2012;Lee et al., 2010) or have limited influences because of other environmental constraints on plant growth (Brookshire and Weaver, 2015). In this study, we deduced future AGB dynamics with both including and not including the effect of $CO_2$ enrichment on grassland AGB. In including $CO_2$ enrichment effect, we used the relationship between $CO_2$ concentration and ANPP based on long-term experimental data derived from Polley et al. (2019). Here, we assumed a general linear response of AGB to increased $CO_2$ concentrations, i.e., an increase of 100 ppm in $CO_2$ leads to an increase of 850 kg ha$^{-1}$ in grassland AGB (Fig. S4b). This linearly positive effect of $CO_2$ on AGB is further applied to the model predicted future AGB (i.e., the AGB not including $CO_2$ enrichment effect). In brief, we used the annual $CO_2$ concentrations under each RCP scenario in the future (Fig. S4a) and the average annual $CO_2$ concentration over 2014-2019 as a baseline, together with the relationship between changes in $CO_2$ and AGB (Fig. S4b), to determine the increment in AGB at

each year from 2020 to 2100. All statistical analyses and graphical productions in this study were performed in R v3.6.3 (R Development Core Team, 2020).

## 3 Results

The field measurements indicate that, on average, aboveground biomass (AGB) in Inner Mongolian grasslands is 1,700 kg ha$^{-1}$ ranging from 220 kg ha$^{-1}$ [2.5% confidence intervals (CI)] to 4,827 kg ha$^{-1}$ (97.5% CI, Fig. 2). Across the three grassland types, meadow steppe has the highest AGB (2,561 Mg ha$^{-1}$ ranging from 736 Mg ha$^{-1}$ to 5,537 Mg ha$^{-1}$), followed by typical steppe (1,496 Mg ha$^{-1}$ ranging from 213 Mg ha$^{-1}$ to 4,418 Mg ha$^{-1}$), and desert steppe has the lowest AGB (835 Mg ha$^{-1}$ ranging from 234 Mg ha$^{-1}$ to 1,928 Mg ha$^{-1}$, Fig. 2).

The fitted three individual machine learning algorithms (i.e., RF, Cubist and SVM) can explain overall 32%-48% of the variance in observed AGB (Fig. 3a, b and c). The ensemble model of the three algorithms can better simulate the observations than any of those individual models (Fig. 3). On average, 52% of the variance in the observations can be explained by the ensemble model (Fig. 3d). Although the variable importance differed among the three algorithms, climatic and livestock variables seem to substantially regulate the AGB dynamics (Fig. S5). After excluding the covariates with high multilinearities, the remaining 10 climatic attributes, 5 edaphic variables and three livestock predictors generally show small autocorrelations (Fig. 4a). Mantel test suggests that, compared to the edaphic and livestock attributes, the climatic variables are in general stronger correlators of AGB in the three grassland types (Fig. 4a). Furthermore, the path analysis suggests that AGB shows small correlations with climate (using the 10 climatic indicators shown in Fig. 4a) and soil (reflected by the five edaphic properties shown in Fig. 4a) while significantly and positively correlates with livestock (Fig. 4b). We also found that climate can indirectly affects AGB via its influence on soil (Fig. 4b). It should be noticed that the small average magnitude with large variabilities of the loadings for climate (Fig. 4b) suggests the corresponding indicators for climate may distinctly affect AGB dynamics. It should also be noted that the overall performance of the fitted path model ($R^2$=0.22, Fig. 4b) in explaining the variability of AGB is much poorer than those of the machine learning models (Fig. 3). This indicates the complex interactions between the environmental drivers in regulating AGB dynamics.

The model simulated average AGB during 1981-2019 (Fig. 5a) and under RCP4.5 (Fig. 5b) and RCP8.5 (Fig. 5c) in the future all show large spatial variations. On average, the regional AGB during the past four decades is 1,438 kg ha$^{-1}$, the corresponding lower and upper limits of the 95% CI is 479 kg ha$^{-1}$, and 2,284 kg ha$^{-1}$, respectively (Fig. 5a). Across grassland types, meadow steppe has the highest average AGB (2,194 Mg ha$^{-1}$ ranging from 1,153 Mg ha$^{-1}$ to 2,631 Mg ha$^{-1}$), followed by typical steppe (1,552 Mg ha$^{-1}$ ranging from 539 Mg ha$^{-1}$ to 2,200 Mg ha$^{-1}$) and desert steppe (893 Mg ha$^{-1}$ ranging from 405 Mg ha$^{-1}$ to 1,341 Mg ha$^{-1}$, Fig. 5a). Spatially, the average coefficient of variation (CV) in the predictions is lowest in meadow steppe (10.5%), followed by desert steppe (14.6%) and typical steppe (21.8%, Fig. 5d). Over 1981-2019, the regional average AGB displayed a decreasing trend (Fig. 6a). Among the three grassland types, the historical changes in AGB (Fig. 6b, c and d) are

in general consistent with that of the total Inner Mongolian grassland AGB (Fig. 6a). Moreover, the long-term field observations also show large inter-annual variabilities in the grassland biomass (Fig. 7) and can support our predicted temporal biomass dynamics at the regional scale (Fig. 6). For example, at four of the six sites, AGB showed a general decreasing trend (Fig. 7).

If the $CO_2$ enrichment effect on AGB is not considered, our predicting results show that future AGB in general decreases under both scenarios of RCPs (i.e., RCP4.5 and RCP8.5, Fig. 6a and Table 2). Compared with the historical AGB (i.e., average AGB during 1981-2019, hereafter the same), on average, AGB at the end of this century (i.e., average of 2080-2100, hereafter the same) would decrease by 14% under RCP4.5 and 28% under RCP8.5, respectively (Table 2). The decreases in AGB under future climate change show large disparities across different grassland types and climate change scenarios. Compared with the

historical average AGB, AGB at the end of this century under RCP4.5 is estimated to decrease by a smaller extent (i.e.,10%) in meadow steep than those in typical (16%) and desert steep (21%, Table 2). In general, AGB under RCP8.5 would reduce by larger extents compared with those under RCP4.5. Under RCP8.5, the average AGB at the end of this century is estimated to experience a 24% (in meadow steep), 30% (in typical steep) and 25% (in desert steep) reduction, compared with the averages over 1981-2019 (Table 2). The magnitudes and spatial patterns of CV in the simulations under both RCP4.5 (Fig. 5e) and

RCP8.5 (Fig. 5f) are comparable with those during the period of 1981-2019 (Fig. 5d).

If the $CO_2$ enrichment effect on AGB is included, the predicted losses in future AGB can be reversed under both RCP scenarios and over different grassland types (Fig. 8). By the end of this century, the regional average AGB is increased by 63% under RCP4.5 and 232% under RCP8.5, respectively, compared with the average AGB during 1981-2019 (Fig. 8a, Table 2). The magnitudes of increases in future AGB differ across different grassland types. For example of RCP4.5, the average AGB at

240 the end of this century is estimated to increase by 40% in meadow steppe, 55% in typical steppe and 102% in desert steppe, respectively, compared with their counterparts during 1981-2019 (Fig. 8b, c and d, Table 2). The increases in AGB are much larger under RCP8.5 than those under RCP4.5. On average, under RCP8.5, the AGB at the end of this century is projected to enhance by 147%, 212% and 394% in meadow, typical and desert steppe, respectively, compared with those over 1981-2019 (Fig. 8b, c and d, Table 2).

**4 Discussion**

Our results, based on AGB observations derived from six long-term field experiments and literature synthesis, indicate the large spatial disparities in aboveground biomass across different grassland types (Fig. 2). This gradient spatial pattern in AGB is comparable with Ma et al. (2008), who carried out a comprehensive field measurements and investigated 113 locations in Inner Mongolian temperate grasslands during 2002-2005. On the regional scale, we mapped grassland AGB at high spatial

resolution, which shows that AGB generally decreases from north-eastern to south-western areas in the study region (Fig. 5a). Such a spatial pattern is also consistent with the maps generated from remote sensing derivations (Fig. S6). This demonstrates

the accuracy of our data-driven predictions on AGB. It should be noted that existing mapping products of grassland AGB use mainly remote sensing approaches requiring inputs from satellite-based datasets (Guo et al., 2016;Jiao et al., 2019;Ma et al., 2010a). Our fitted machine learning model, however, uses only several readily obtainable environmental covariates (Fig. 4 and

Table 1). Our results demonstrate the ability of machine learning approach to effectively extrapolate grassland AGB to much larger spatiotemporal extents (e.g., Fig. 5 and 6).

Our simulation results show that, under the climate warming over the past four decades (Fig. S3), the average AGB generally experienced a declining trend in the study region (Fig. 6). This may partly supports the possible negative effects of temperature rising on AGB that has been widely reported (De Boeck et al., 2008;Wang et al., 2020a), particularly in the arid and semi-arid

ecosystems (Ma et al., 2010b). This harmful influence of warming on AGB is explainable. For example, in a system restrained by water availability (e.g., temperate grassland), warming can not only inhibit plant photosynthesis (Xu and Zhou, 2005) but also enhance evaporation and further intensify water stress (De Boeck et al., 2006) thereby decreasing grassland biomass. Precipitation has generally been recognized to have positive effects on AGB in the temperate grasslands (Hovenden et al., 2019;Ma et al., 2010a), which supports our findings in this study. For example, the simulated average AGB is relatively higher

in the years with higher MAP (e.g., 1998 and 2012) than those in other years (Fig. 6a). The importance of precipitation on AGB can be more reflected by the spatial patterns of these two attributes, e.g., AGB is much lower in the more arid regions (Fig. 5a) where soils are suffering severer water deficiencies. Apart from climatic factors, our results also demonstrate the co-regulating effects of soil conditions and livestock on the dynamics of grassland AGB as indicated by the machine learning models (Fig. S5) and the path analysis model (Fig. 4b). For example, the increasing trend in livestock over the past four decades

(Fig. S2d) is generally in line with the overall decreasing trend in the contemporary AGB (Fig. 6a). It should be noted that the major drivers of the simulated temporal changes in AGB (Fig. 6) can vary during different periods in this study due to data unavailability particularly for livestock. Specifically, AGB dynamics over 1981-2019 is co-regulated by both changes in climates and livestock (Fig. S2, S3 and S5). In future scenario simulations (e.g., 2020-2100, Fig. 6), however, AGB variations are predominantly controlled by climates since a constant grazing intensity was adopted over time in future predictions (see

Materials and Methods). We admit that the actual grazing intensity can vary over time in the future under different RCP scenarios and simply assuming a stable grazing intensity over time can lead to substantial biases in AGB estimations. We need novel approaches to derive the temporal variations in grazing intensity at larger temporal extents.

Our estimations indicate that AGB can be substantially increased under future $CO_2$ enrichment (Fig. 8). Here, several uncertainties and limitations should be noticed in interpreting our results. First, the gradient of $CO_2$ concentrations in Polley

et al. (2019), which is used to derive the effect of $CO_2$ enrichment on AGB, has a smaller range (i.e., 250 ppm to 500 ppm) than those under RCP8.5 [i.e., around 900 ppm by the end of this century, Fig. S4a]. Here, extrapolations of such a relationship between $CO_2$ concentration and AGB to larger extents of $CO_2$ concentrations can lead to substantial uncertainties in estimations of AGB. Second, the local soil (Fay et al., 2012) and climatic (Brookshire and Weaver, 2015) factors can modify the actual

CO$_2$ enrichment effect on AGB, which may also result in large uncertainties in the quantified AGB. For example, any
stimulation in plant growth is constrained by the availability of other resources required by plant growth (Reyes-Fox et al.,
2014) such as soil water availability (Brookshire and Weaver, 2015). Consequently, the magnitude of the increases in AGB
induced by CO$_2$ enrichment estimated in this study, particularly under RCP8.5, can be largely overestimated due to possible
deficiencies of either nutrients or water required by plant growth (Wang et al., 2020b).

We also notice that our model predictions show larger inter-annual variations in AGB (Fig. 6a) than those in the estimations
based on remote-sensing approaches (Fig. S6). In fact, the remote sensing derived AGB has also been bias-corrected by the
field measurements (Jiao et al., 2019). Consequently, this disparity could be related to the difference of observed AGB datasets
used in different studies. Specifically, the measurements of biomass used to calibrate remote-sensing data [normalized
difference vegetation index (NDVI)] in Jiao et al. (2019) were generally conducted during 2001-2015. Extrapolations of these
observations from a short term (e.g., 2001-2015) to a much longer term (e.g., 1982-2015) might lead to underestimations in
the long-term interannual variabilities. Our study, however, integrate the *in situ* observed data from six long-term (1982-2015)
field experiments (Fig. 1), which can potentially better represent the AGB over larger temporal scales. It is noteworthy that
the accuracy of our predictions on future grassland AGB relies substantially on the robustness of future climate change
projections simulated by the GCMs (e.g., CESM1-BGC). However, although CESM1-BGC (like all the other CMIP5 models)
can reasonably well simulate changes in temperature, it may not well predict precipitation, particularly for Eastern China that
affected by large-scale atmospheric circulations (Huang et al., 2013). In addition, the effects of solar radiation (Yu et al.,
2021;Zhang et al., 2020a) and its complex responses to dust aerosol (Fu et al., 2009;Qi et al., 2013;Wang et al., 2013) on plant
photosynthesis and biomass formation were not considered in this study, which can be another source of uncertainties in the
estimated AGB under future climate change. Last but not least, the assumption of space-for-time substitution has been widely
debated and challenged (Johnson and Miyanishi, 2008;Walker et al., 2010). Although grassland type across space is treated as
an independent predictor of AGB in this study, we admit that using the spatial gradients of observations to predict AGB
backward or forward in time may still lead to large uncertainties. Consequently, it should be cautious in interpreting the
modelled future grassland AGB in this study.

## 5 Conclusions

Our results demonstrate that the aboveground biomass in Inner Mongolian grasslands shows large spatial and temporal
variations during the past four decades, which is driven by a series of environmental covariates. Particularly, current climate
change characterized mainly by warming together with an increased grazing intensity can have negative effects on grassland
AGB. The decreases of AGB, however, can potentially be reversed by the positive effects of atmospheric CO$_2$ enrichment. In
addition, our results demonstrate that adopting a machine learning model approach with only a few readily obtainable
environmental predictors can accurately capture AGB dynamics, which enables extrapolations of AGB across larger

spatiotemporal extents. Moreover, our study provides new data on annual AGB in Inner Mongolian grasslands at fine spatial (1km) and temporal (yearly) resolutions at large temporal scales (1981-2100).

**Data availability.** The data that support the findings of this study (Data S1) are openly available at: 10.6084/m9.figshare.13108430.

**Supplement.** The supplement related to this article is available online at: XXX.

**Author contributions.** G. Wang and Y. Huang conceived this study. G. Wang conducted the data analysis with interpretations from Z. Luo and Y. Huang. G. Wang and Z. Luo prepared the article with contributions from all authors.

**Competing interests.** The authors declare that they have no conflict of interest.

**Acknowledgements.** The authors acknowledge the people who conducted the filed long-term experiments and collected the observed data.

**Financial support.** This study is financially supported by the National Natural Science Foundation of China (Grant No. 41775156 and 41590875) and the Strategic Priority Research Program of the Chinese Academy of Sciences (Grant No. XDA26010103).

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

**Table 1 The environmental covariates used in this study.**

| Covariates | Code | Description | Unit |
|---|---|---|---|
| Edaphic variables | CFRAG | Coarse fragments (>2mm) | % |
| | BULK | Bulk density | $g\ cm^{-3}$ |
| | ORGC | Organic carbon | $g\ kg^{-1}$ |
| | SDTO | Sand content | % |
| | CLPC | Clay content | % |
| | STPC | Silt content | % |
| | TAWC | Available water capacity | $cm\ m^{-1}$ |
| | TOTN | Total nitrogen | $g\ kg^{-1}$ |
| | CNrt | C:N ratio | - |
| | PHAQ | pH measured in $H_2O$ | - |
| Climatic variables | T1 | Annual mean temperature | °C |
| | T2 | Mean diurnal range | °C |
| | T3 | Isothermality (T2/T7×100) | % |
| | T4 | Temperature seasonality (standard deviation×100) | °C |
| | T5 | Max temperature of warmest month | °C |
| | T6 | Min temperature of coldest month | °C |
| | T7 | Temperature annual range (T5–T6) | °C |
| | T8 | Mean temperature of wettest quarter | °C |
| | T9 | Mean temperature of direst quarter | °C |
| | T10 | Mean temperature of warmest quarter | °C |
| | T11 | Mean temperature of coldest quarter | °C |
| | P1 | Annual precipitation | mm |
| | P2 | Precipitation of wettest month | mm |
| | P3 | Precipitation of driest month | mm |
| | P4 | Precipitation seasonality (coefficient of variation) | % |
| | P5 | Precipitation of wettest quarter | mm |
| | P6 | Precipitation of driest quarter | mm |
| | P7 | Precipitation of warmest quarter | mm |
| | P8 | Precipitation of coldest quarter | mm |
| | MATG | Mean annual temperature during growing season | °C |
| | MATNG | Mean annual temperature during non-growing season | °C |
| | MAPG | Mean annual precipitation during growing season | mm |
| | MAPNG | Mean annual precipitation during non-growing season | mm |
| Grassland type | - | Meadow, typical and desert steppe | - |
| Livestock | - | Cattle, sheep and goat | $head\ km^{-2}$ |

**Table 2 Summary of Inner Mongolian grassland aboveground (AGB) biomass during different periods**

| $CO_2$ enrichment effects | Climate change scenario | Period | AGB across grassland types (kg ha$^{-1}$, mean±SD) | | | |
|---|---|---|---|---|---|---|
| | | | Meadow | Typical | Desert | All |
| Not included | RCP4.5 | 2020−2039 | 1,934±112 | 1,345±201 | 918±287 | 1,304±181 |
| | | 2040−2059 | 1,837±171 | 1,223±235 | 768±340 | 1,174±249 |
| | | 2060−2079 | 1,916±117 | 1,312±184 | 779±275 | 1,253±191 |
| | | 2080−2100 | 1,965±97 | 1,306±170 | 702±279 | 1,237±181 |
| | RCP8.5 | 2020−2039 | 1,902±107 | 1,269±156 | 740±294 | 1,206±163 |
| | | 2040−2059 | 1,862±142 | 1,230±245 | 733±304 | 1,165±252 |
| | | 2060−2079 | 1,800±123 | 1,219±193 | 722±308 | 1,169±202 |
| | | 2080−2100 | 1,672±140 | 1,087±156 | 666±236 | 1,033±162 |
| Included | RCP4.5 | 2020−2039 | 2,187±161 | 1,597±224 | 1,171±340 | 1,557±220 |
| | | 2040−2059 | 2,520±199 | 1,906±264 | 1,451±346 | 1,857±272 |
| | | 2060−2079 | 2,919±143 | 2,315±217 | 1,782±295 | 2,256±223 |
| | | 2080−2100 | 3,067±103 | 2,408±172 | 1,804±283 | 2,339±184 |
| | RCP8.5 | 2020−2039 | 2,274±166 | 1,642±176 | 1,113±307 | 1,579±177 |
| | | 2040−2059 | 3,012±261 | 2,380±314 | 1,882±345 | 2,315±310 |
| | | 2060−2079 | 4,097±331 | 3,517±351 | 3,018±471 | 3,466±360 |
| | | 2080−2100 | 5,423±470 | 4,838±503 | 4,417±585 | 4,784±512 |

465

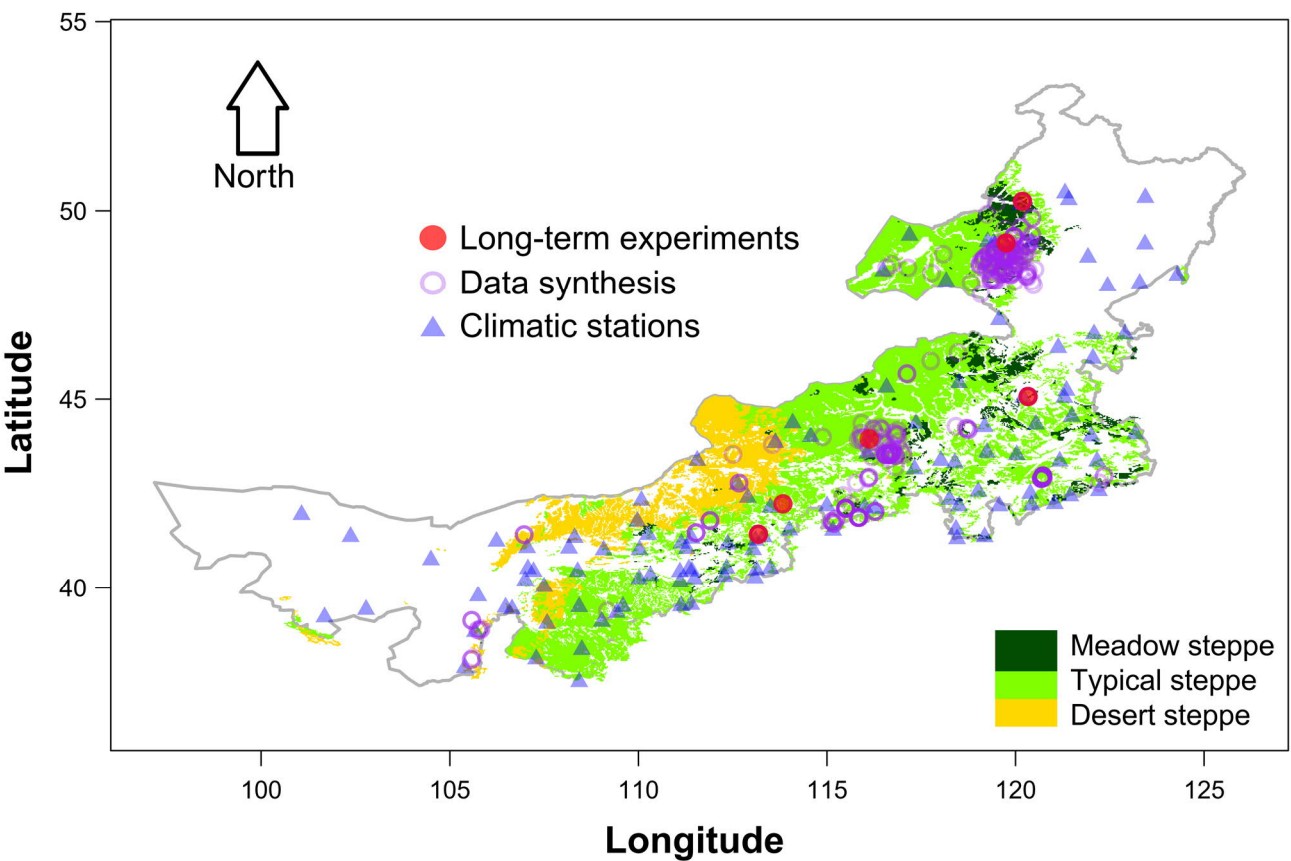

**Figure 1. Spatial distribution of grassland aboveground biomass observations and climatic stations in Inner Mongolia.** The Inner Mongolian grasslands are grouped into three categories (i.e., meadow steppe, typical steppe and desert steppe). Observations of grassland biomass were both derived from the six long-term experimental sites and data synthesis of existing studies. The ground climatic records were obtained from National Meteorological Information Centre (NMIC) of China.

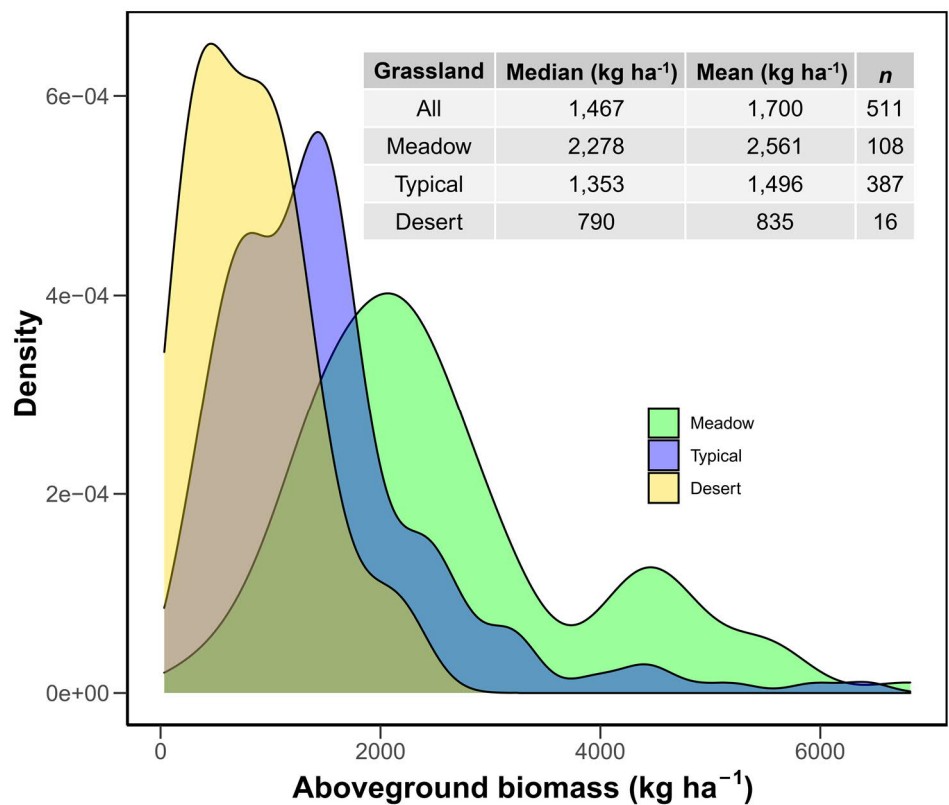

**Figure 2. Aboveground biomass distribution across different grassland types in Inner Mongolia.** See Fig. 1 for the spatial distribution
of the three grassland types in Inner Mongolia.

475

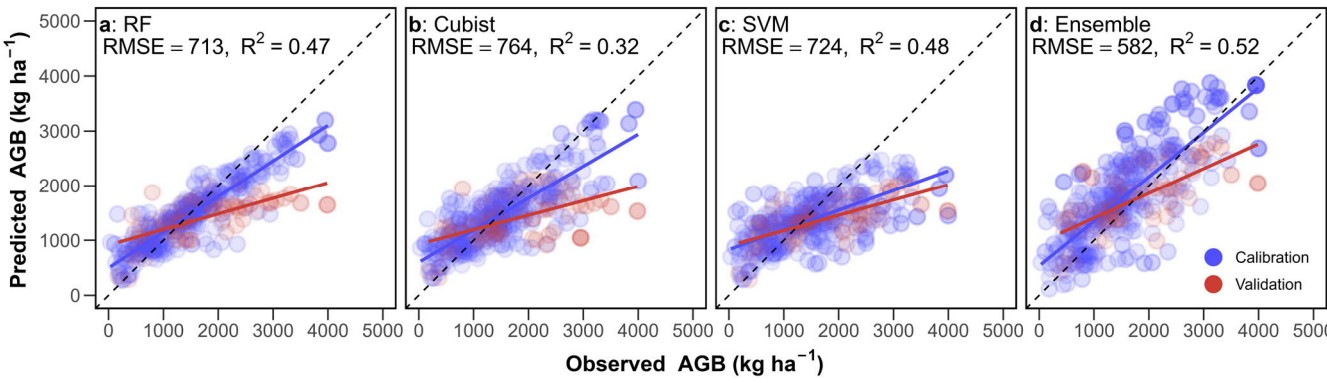

**Figure 3. Performances of models to predict grassland aboveground biomass (AGB).** a, random forest (RF); b, Cubist; c, support vector machines (SVM); d, the ensemble model of a-c. For each individual model, 80% of the stratified samples of observations were used for model calibration, with the other 20% used for validation. $R^2$ and RMSE show the coefficient of determination and root mean square error of model validations. In model calibrations, the $R^2$ is 0.82, 0.66 and 0.43 for RF, Cubist and SVM, respectively, and RMSE is 359 kg ha$^{-1}$, 460 kg ha$^{-1}$ and 579 kg ha$^{-1}$, respectively for RF, Cubist and SVM, respectively.

480

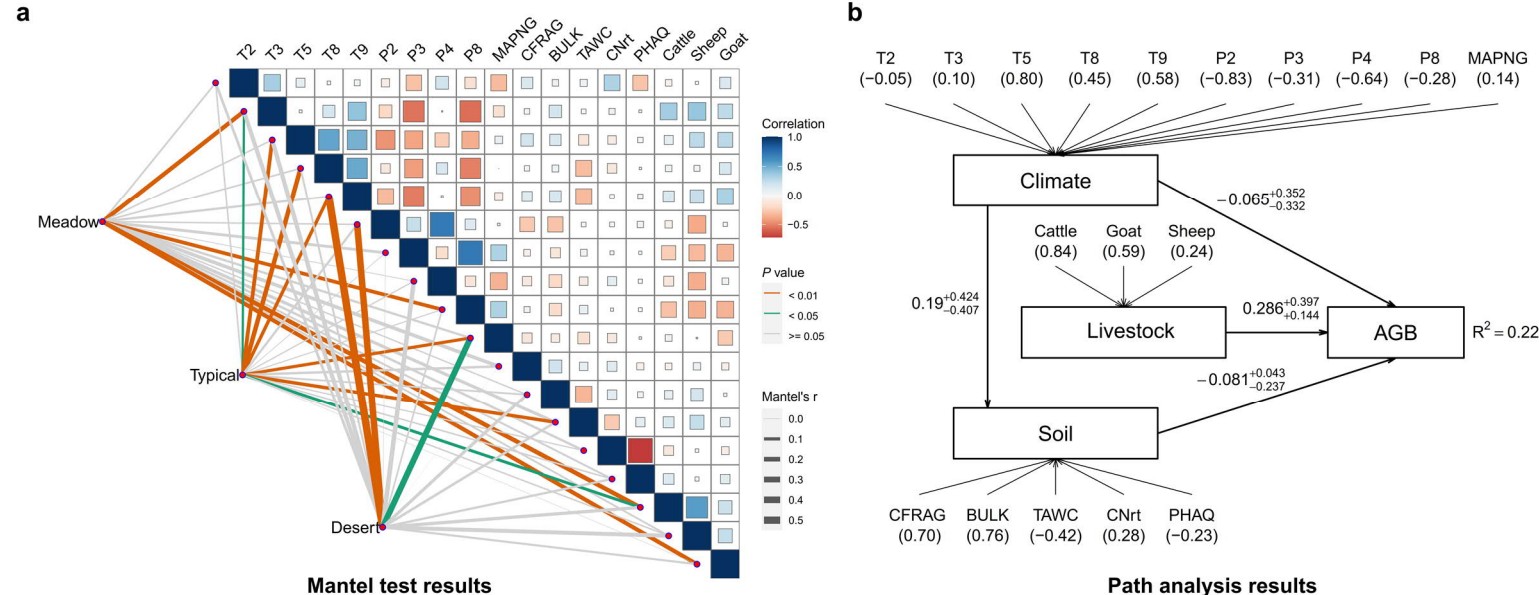

**Figure 4. Environmental drivers of aboveground grassland biomass (AGB).** a, the correlation matrix of environmental drivers and mantel test results. The upper triangle shows the pairwise comparisons of predicting variables, with a color gradient denoting Spearman's correlation coefficient. Taxonomic grassland type (i.e., meadow, typical and desert steppe) was related to each environmental factor by partial (geographic distance–corrected) Mantel test. Line color represents the statistical significance and line width denotes the Mantel's r statistic for the corresponding distance correlations. b, the path analysis results of the direction and magnitude of the effects of latent variable climate (reflected by T2, T3, T5, T8, T9, P2, P3, P4, P8 and MAPNG), soil (using CFRAG, BULK TAWC, CNrt and PHAQ as indicators) and livestock (using Cattle, Goat and Sheep as indicators) on AGB. Numbers in parentheses represent the loadings (correlation coefficients) of the indicators to the latent variables. See Table 1 for descriptions of each variables and see details in Materials and Methods section for the statistical analysis.

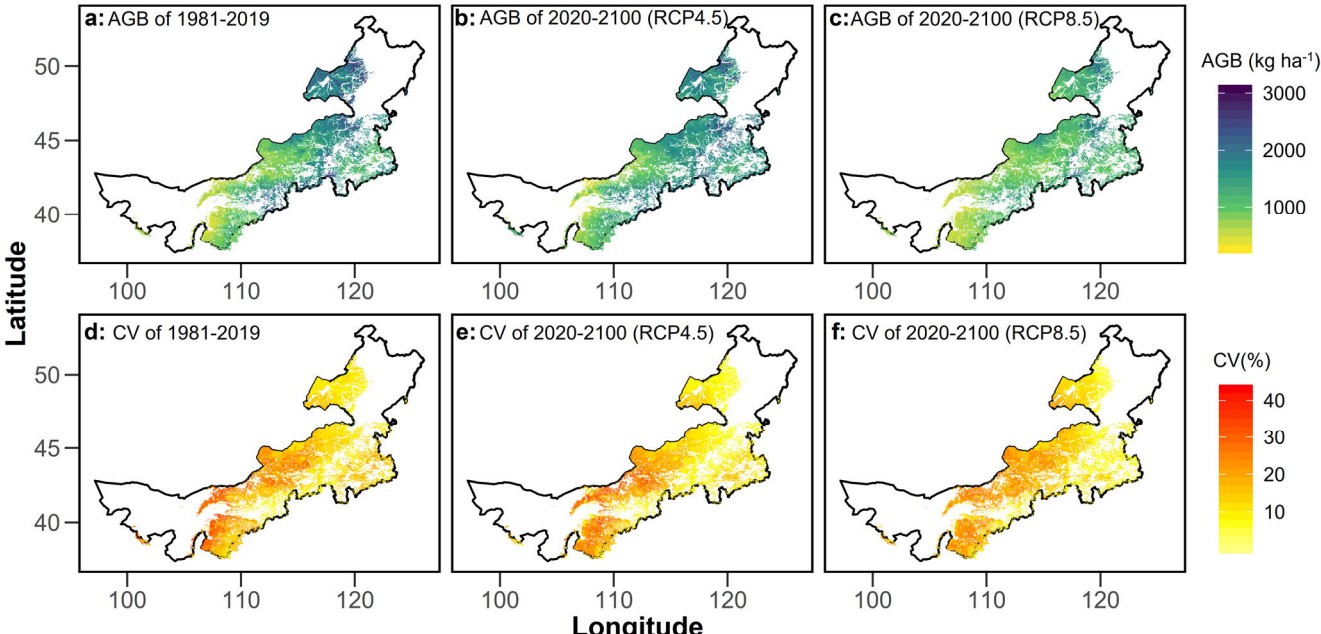

**Figure 5. Spatial patterns of Inner Mongolian grassland aboveground biomass (AGB) and the uncertainties in terms of coefficient of variations (CV).** The upper panel shows the average gridded AGB over 1981-2019 (a) and under two climate change scenarios [RCP4.5 (b) and RCP8.5 (c)] over 2020-2100. The lower panel (d, e and f) exhibit the associated CV of the upper panel. Please note that these estimations were derived from simulations without considering the atmospheric $CO_2$ enrichment effects on AGB.

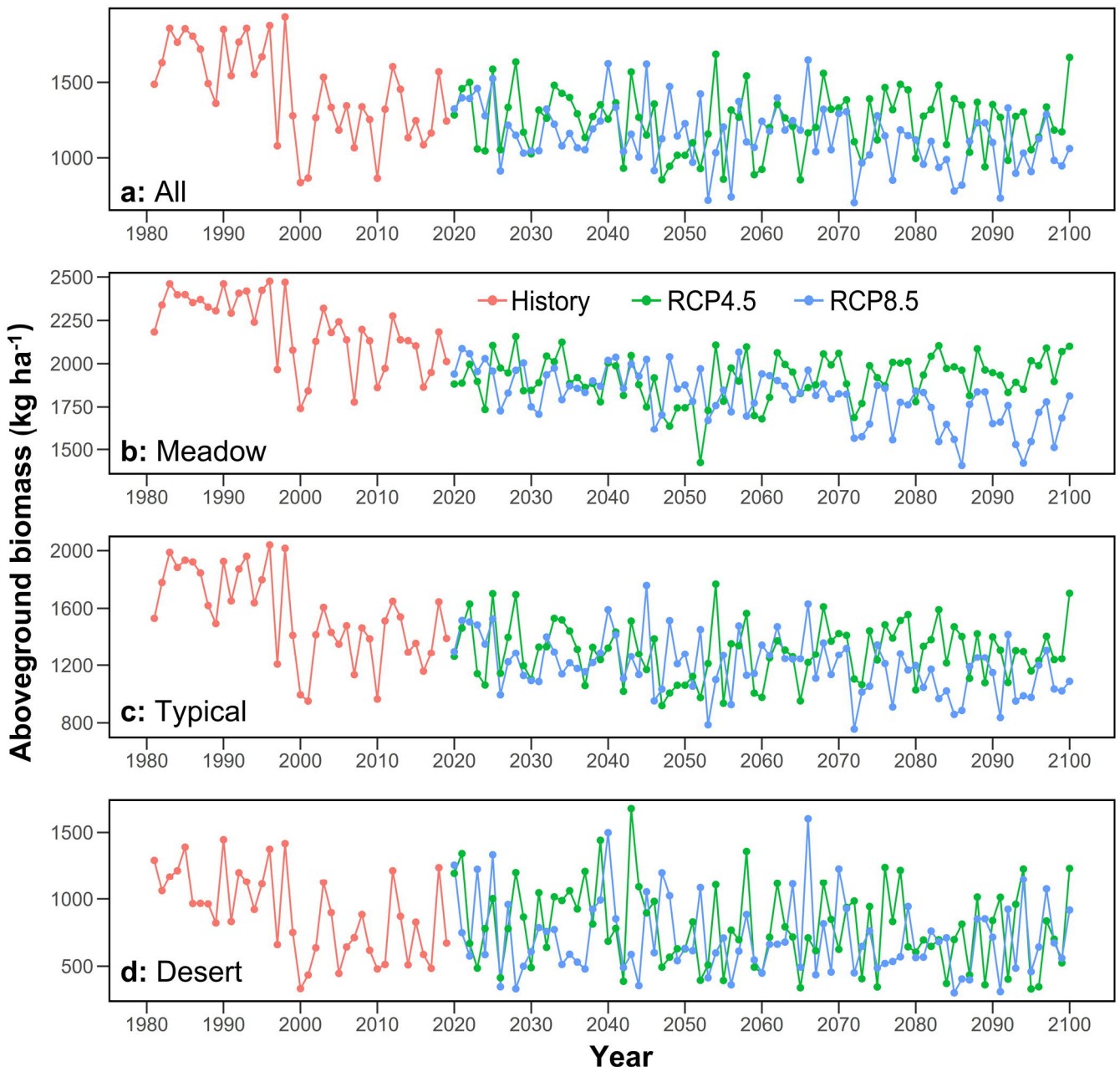

**Figure 6. Temporal variations of the predicted average aboveground biomass (AGB) in Inner Mongolian grasslands.** At each year, data are averages of all the 1km×1km grids (a) and across a certain grassland type at the regional scale (b, c and d). It should be noticed that these estimations were derived from simulations without considering the atmospheric $CO_2$ effects on AGB.

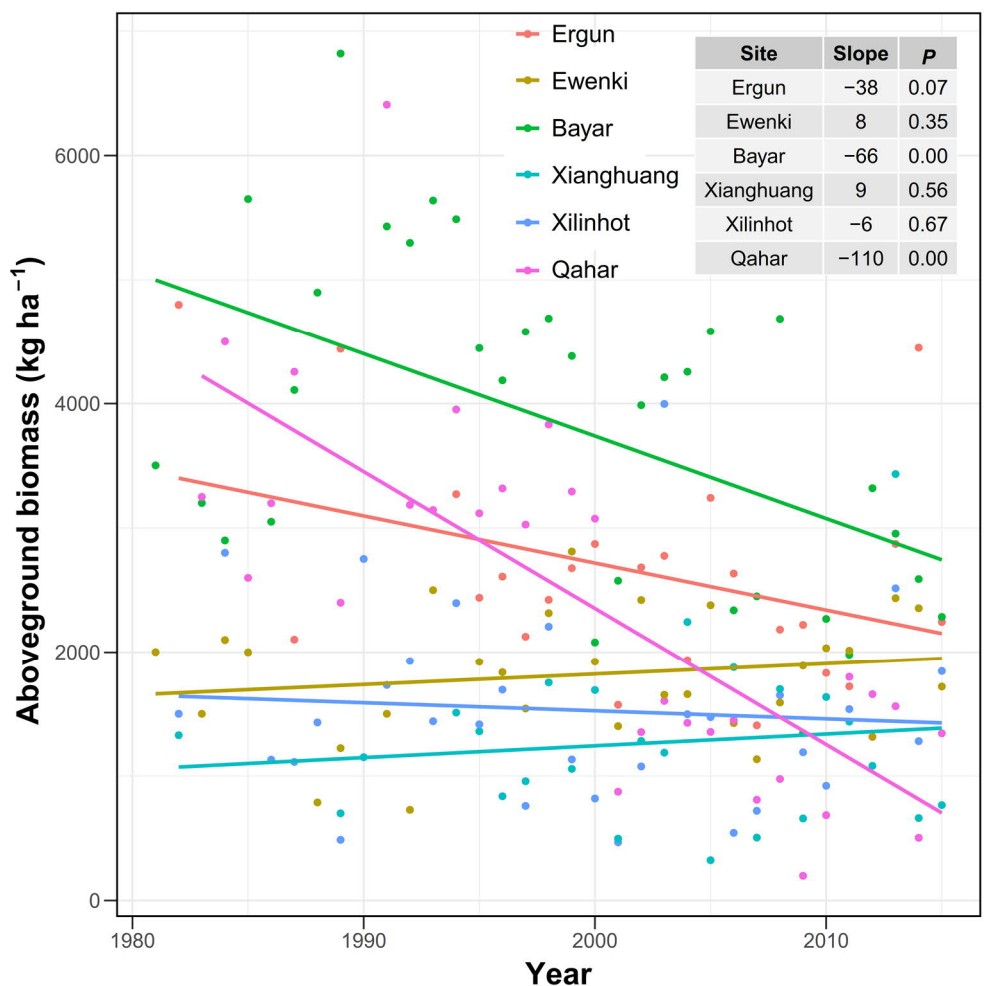

**Figure 7. Temporal changes in aboveground biomass (AGB) in the six long-term filed experiments in Inner Mongolian grasslands.** The table inside shows the linear trends (slope, kg ha[-1] yr[-1]) in AGB and the significances (reflected by *P* value).

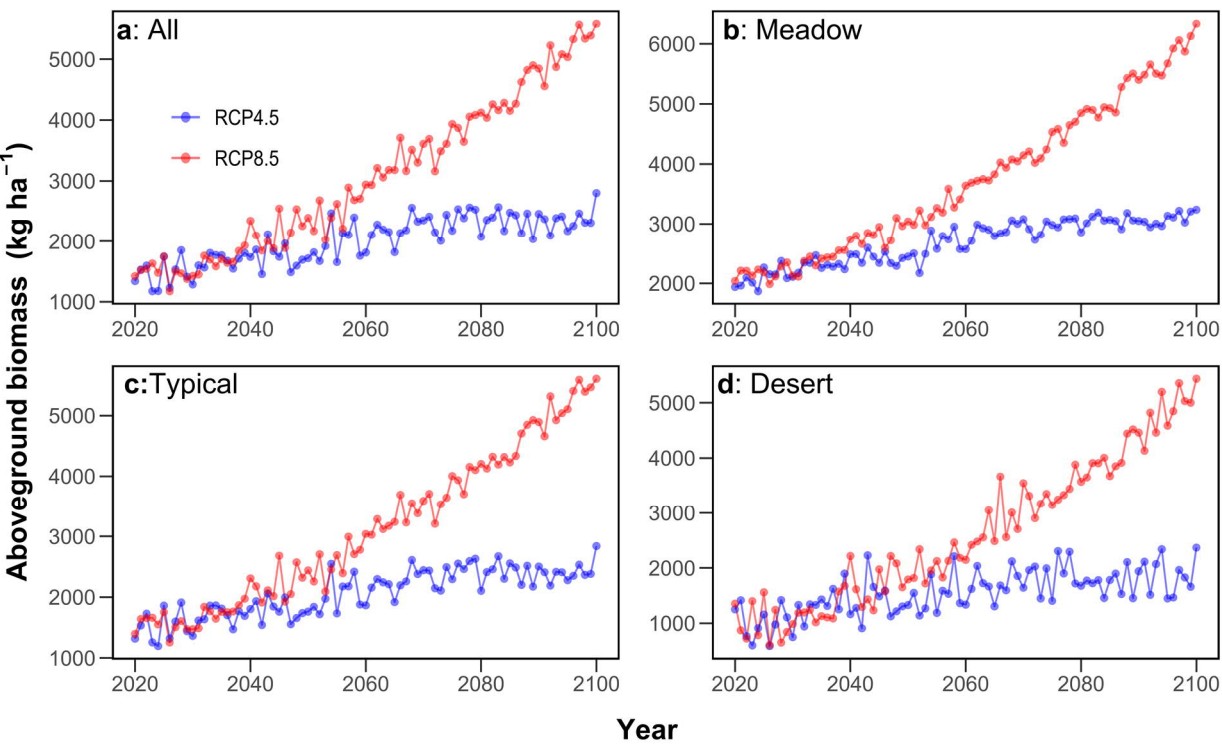

505

**Figure 8. Estimated future aboveground biomass (AGB) in Inner Mongolian grasslands when the $CO_2$ enrichment effects on AGB is considered.** The temporal changes in AGB of all Inner Mongolian grasslands (a), meadow (b), typical (c) and desert (d) steppe are presented in different sub-figures.