# Peer review of "Aboveground biomass in Inner Mongolian temperate grasslands decreases under climate warming"

_Atmospheric Chemistry and Physics, 2020_

## Referee Comment (RC1) · Anonymous Referee #1 · 4 Dec 2020

Review: ACP-2020-1088

The work by Wang and co-authors viewed the above-ground biomass (AGB) over Inner Mongolian temperate grasslands, and analyzed the relationships of many climate, soils, grazing intensity, and grassland type variables to plants using combine biomass measurements from six long-term experiments and data in existing literatures. They found that under future climate warming, AGB in the study region could continue to decrease. On average, compared with the historical AGB (i.e., average of 1981-2019), the AGB at the end of this century (i.e., average of 2080-2100) would decrease by 14% under RCP4.5 and 28% under RCP8.5, respectively. The paper is of interests to the broad readership of Atmospheric Chemistry and Physics. The finding may also help advance our understanding how global climate change influence the temperate grass-

lands. Yet there are several limitations in the current version that need to be addressed before the publication.

I. There are several concepts in the manuscript that are not easy to understand, including 'temperate grasslands', 'meadow steppe', 'typical steppe', and 'desert steppe'.

II. Introduction. The novelty of your work should be emphasized and explained in a better way.

III. Materials and Methods. Lines 125-126, identifying the root mean square error (RMSE) is not clear here, please add its calculations (equations) and units (% or kg ha-1?).

IV. Results. Lines 172-175, that is, Figure 4. Correlation matrix cannot indicate environmental drivers of Inner Mongolian grassland biomass. Thus, maybe you could employ that correlation matrix combined with structural equation modelling analysis of the environmental factors effect on AGB.

V. In that way, in Supplementary Figures S2-S4, AGB in Inner Mongolian temperate grasslands decreases, who is major driver? Climate change (e.g., temperature) or human activities (e.g., grazing intensity)?

VI. Lines 182-184, i.e., Figure 6. You need to increase the segmentation fitting lines to ensure the description more clearly in Figure 6s.

VII. Discussion. Lines 219-223, I do not understand the argument that's being made here. I think, a better work could be finished to set up the questions here in the rest of the introduction-talking more about how climate warming (major driver?) linked to soil conditions and livestock (positive feedback?) might affect AGB, and so on?

Minor comments:

1. P2, L41-42, L49, L57, in many cases, citations of references are not arranged systematically. Either it should be chronologically or alphabetically arranged.

[Figure]

2. P14, L364-367, model calibration (80% samples) does not need to enumerate R2 and RMSE? Why the proportion for model calibration and validation is 80%:20%? Why not 50%:50%?

I wish the above suggestions or comments can help improve the quality of this manuscript. Thanks in advance.

---

## Referee Comment (RC2) · Anonymous Referee #2 · 12 Dec 2020

The authors compiled in situ measurements and long-term experimental data to estimate changes in aboveground biomass over Inner Mongolia at a spatial resolution of 1 km. Moreover, the machine-learning model which was constructed using historical observations is applied to estimate aboveground biomass changes under future climate scenarios. Without implicitly considering the following two major comments, I would not recommend this paper to be published. In a warmer future, the rising $CO_2$ effect on aboveground grassland productivity was not considered. It is well established that the fertilization effect of rising $CO_2$ would greatly offset the warming-induced productivity loss in grasslands. Without considering the $CO_2$ effect, projections of aboveground biomass would be greatly biased in a warmer world. Note that the temporal dynamics was deduced from the analysis of climate drivers of spatial gradient in aboveground

biomass. This space-for-time method was generally challenged by the fact that the climatic controls in space and time would be different. The authors have six long-term experimental sites, but unfortunately these valuable data set especially for evaluating the inferred long-term trend has not been explored. The authors should add new analyses and figures to evaluate the model-derived productivity changes in terms of mean, inter-annual variation and trend using these data. Minor comments: 1. Change the error of AGB unit "ka ha-1" in Abstract Line 17. 2. A table showing the details of environmental drivers might be helpful. 3. In the future projection, current grazing intensity was kept stable, while it would not be consistent with RCP simulations under RCP4.5 and RCP8.5. Some hypothetic scenarios are necessary

---

## Author Comment (AC1) · 3 Jan 2021

Reviewer #1: The work by Wang and co-authors viewed the above-ground biomass (AGB) over Inner Mongolian temperate grasslands, and analyzed the relationships of many climate, soils, grazing intensity, and grassland type variables to plants using combine biomass measurements from six long-term experiments and data in existing literatures. They found that under future climate warming, AGB in the study region could continue to decrease. On average, compared with the historical AGB (i.e., average of 1981-2019), the AGB at the end of this century (i.e., average of 2080-2100) would decrease by 14% under RCP4.5 and 28% under RCP8.5, respectively. The paper is of interests to the broad readership of Atmospheric Chemistry and Physics. The finding may also help advance our understanding how global climate change influence

the temperate grasslands. Yet there are several limitations in the current version that need to be addressed before the publication.

Authors' Response: We greatly appreciate the reviewer's positive comments and understandings of our study.

There are several concepts in the manuscript that are not easy to understand, including 'temperate grasslands', 'meadow steppe', 'typical steppe', and 'desert steppe'.

Authors' Response: Thanks to the reviewer's comments. The study region (i.e., Inner Mongolian grasslands) is characterized mainly by a temperate climate (Zhang et al., 2020) and thus is also called Inner Mongolian temperate grassland. The grasslands in the study region can be generally classified into three categories, i.e., meadow steppe, typical steppe and desert steppe (National Research Council, 1992). In brief, meadow steppe is distributed mainly in the eastern steppe zones, typical steppe locates mostly in the central Inner Mongolia, and desert steppe is found mainly to the west of the typical steppe (Fig. 1). We have clarified this information in the revised MS (Line 83 – 87).

Introduction. The novelty of your work should be emphasized and explained in a better way.

Authors' Response: Thanks to the reviewer's suggestions. In the revision, we have modified the Introduction section in a more logic flow (Line 41-45; 53-55; 63-72; 79-80). Specifically, we have emphasized that we aim to explicitly take into account the seasonality of climate, soil, grassland type and grazing intensity in assessing the spatiotemporal variations of AGB. Moreover, we also highlight that we aim to predict the future AGB dynamics under climate change characterized mainly by warming. The possible promoting effect of $CO_2$ enrichment on AGB is also included in the revision as recommended by another reviewer and emphasized in the Introduction. Please see details in the Introduction of the revision, we hope these modifications can satisfy the concerns of the reviewer.

Materials and Methods. Lines 125-126, identifying the root mean square error (RMSE) is not clear here, please add its calculations (equations) and units (% or kg ha-1?).

Authors' Response: Added accordingly (Line 141-143).

Results. Lines 172-175, that is, Figure 4. Correlation matrix cannot indicate environmental drivers of Inner Mongolian grassland biomass. Thus, maybe you could employ that correlation matrix combined with structural equation modelling analysis of the environmental factors effect on AGB.

Authors' Response: Thanks to the reviewer for her/his constructive suggestions. We have added a path analysis on relationship between the environmental factors and AGB in the revised MS (Line 154-161, 210-218, Fig. 4b). Specifically, the path modelling analysis suggests that AGB shows small correlations with climate (using the 10 climatic indicators identified by the analysis to exclude the environmental covariates with high multicollinearities, hereafter the same for soil) and soil (reflected by the five edaphic properties) while significantly and positively correlates with grazing (Fig. 4b). We also found that climate can indirectly affects AGB via its influence on soil (Fig. 4b). It should be noticed that the small average magnitude with large variabilities of the loadings for climate (Fig. 4b) suggests the corresponding indicators for climate may distinctly affect AGB dynamics. It should also be noted that the overall performance of the fitted path model ($R^2=0.22$, Fig. 4b) in explaining the variability of AGB is much smaller than those of the machine learning models (Fig. 3), which indicates that more complex and non-linear relationships of the environmental drivers may exist in regulating AGB dynamics. We have clarified these in the revision (Line 154-161, 210-218, Fig. 4b).

In that way, in Supplementary Figures S2-S4, AGB in Inner Mongolian temperate grasslands decreases, who is major driver? Climate change (e.g., temperature) or human activities (e.g., grazing intensity)?

Authors' Response: The major drivers of the simulated temporal changes in AGB (Fig. 6) can vary during different periods due to data availability particularly for grazing intensity. For example, AGB dynamics over 1981-2019 is co-regulated by both changes in climates and grazing activities (Fig. S2, S3 and S5) and grazing intensity has a higher influence on AGB dynamics (Fig. 4b). In the future scenario simulations (e.g., 2020-2100, Fig. 6), however, AGB dynamics are predominantly controlled by climate changes since a constant grazing intensity was adopted over time in the future predictions. We admit that the actual grazing intensity can vary over time in the future depending on RCP scenarios, simply assuming a stable grazing intensity over time can lead to substantial biases in AGB estimations. We have clarified these in the revised MS (Line 277-284).

Lines 182-184, i.e., Figure 6. You need to increase the segmentation fitting lines to ensure the description more clearly in Figure 6s.

Authors' Response: We have updated the Fig. 6 by more clearly presenting the temporal variations in AGB. Besides, we have also revised the MS by more clearly describing the results associated with Fig. 6 (Line 225-227).

Discussion. Lines 219-223, I do not understand the argument that's being made here. I think, a better work could be finished to set up the questions here in the rest of the introduction-talking more about how climate warming (major driver?) linked to soil conditions and livestock (positive feedback?) might affect AGB, and so on?

Authors' Response: We have clarified these sentences in the revision (Line 273-277). Briefly, we intend to inform that, apart from climatic factors, soil conditions and livestock also co-regulate the dynamics of grassland AGB, which is indicated by the machine learning models (Fig. S4) and the path analysis model (Fig. 4b). Such findings have seldom been assessed on large scales in the study region. These findings are consistent with several findings highlighting the importance of soil physical and chemical characteristics (Griffiths et al., 2012;Yang et al., 2009) and grazing intensity (Eldridge and Delgado‐Baquerizo, 2017) in controlling grassland biomass changes. As mentioned above, we have also summarized that the major drivers of the simulated

temporal changes in AGB (Fig. 6) can vary during different periods due to data availability particularly for grazing intensity (Line 277-284). We hope these clarifications can satisfy the reviewer's concerns.

Minor comments: P2, L41-42, L49, L57, in many cases, citations of references are not arranged systematically. Either it should be chronologically or alphabetically arranged.

Authors' Response: In the revision, we have updated all the citations by alphabetically arranging the references in the main text. P14, L364-367, model calibration (80% samples) does not need to enumerate R2 and RMSE? Why the proportion for model calibration and validation is 80%:20%? Why not 50%:50%?

Authors' Response: We have added the R2 and RMSE for the model calibrations in Figure 3 (Line 471-472). For the proportion of train-test split, there is no universal or best split option, however, the representativeness of both train set and test set is required. In this study, we used the stratified to ensure the dataset representativeness. A training set with the percentage size of 80% and a remainder percentage of 20% for testing set (e.g., that in this study) is one of the most commonly used split percentages in machine learning approaches (Brownlee, 2016). We have clarified this in the revision (Line 137-138).

I wish the above suggestions or comments can help improve the quality of this manuscript. Thanks in advance.

Authors' Response: We highly appreciate the reviewer's constructive comments, which significantly contribute to the improvement of this study.

---

## Author Comment (AC2) · 3 Jan 2021

Reviewer #2: The authors compiled in situ measurements and long-term experimental data to estimate changes in aboveground biomass over Inner Mongolia at a spatial resolution of 1 km. Moreover, the machine-learning model which was constructed using historical observations is applied to estimate aboveground biomass changes under future climate scenarios. Without implicitly considering the following two major comments, I would not recommend this paper to be published. In a warmer future, the rising CO2 effect on aboveground grassland productivity was not considered. It is well established that the fertilization effect of rising CO2 would greatly offset the warming-induced productivity loss in grasslands. Without considering the CO2 effect, projections of aboveground biomass would be greatly biased in a warmer world.

[Figure]

Authors' Response: We greatly appreciate the reviewer's useful suggestions and constructive comments, following which we have substantially revised our MS particularly on the $CO_2$ enrichment effect. We in general stand with the reviewer on her/his opinion that fertilization effect of rising $CO_2$ would greatly offset the warming-induced productivity loss in grasslands. In the revision, first, we derived the relationship between $CO_2$ concentration and ANPP based on the data derived from Polley et al. (2019) (Line 184-191; Fig. S4). Second, by applying this relationship on future $CO_2$ concentrations and AGB projected by the machine learning models under different RCP scenarios, we found AGB losses due to climate change (not including $CO_2$ enrichment effect) can not only be offset but also be reversed (Line 240-248; Fig. 8). Third, we noticed that $CO_2$ enrichment effect on AGB can be dependent on resource availability of other environmental factors such as nutrient and water (Brookshire and Weaver, 2015;Wang et al., 2020), thus there remain large uncertainties in the estimated AGB variations under a rising $CO_2$ as estimated in this study. We have thoroughly discussed these possible uncertainties and limitations of our results (Line 284-294). We hope these revisions can satisfy the reviewer's concerns.

Note that the temporal dynamics was deduced from the analysis of climate drivers of spatial gradient in aboveground biomass. This space-for-time method was generally challenged by the fact that the climatic controls in space and time would be different.

Authors' Response: We respectfully disagree with the reviewer on this point although we can understand her/his potential concerns, which is not the case in our study. In this study, we split the study region into three categories (e.g., meadow steppe, typical steppe and desert steppe), these three categorical variables are included as predictors in the machine learning models as dummy variables. This is to avoid deducing the dependent variables in a certain category using the independent variables (e.g., climate variables) across other categories in building the machine learning models, i.e., predicting an apple using an orange. Consequently, we have realized that the climatic controls over spaces are different and they have actually already been taken into account in building the machine learning models. We have further clarified this in the revised MS (Line 134-136).

The authors have six long-term experimental sites, but unfortunately these valuable data set especially for evaluating the inferred long-term trend has not been explored. The authors should add new analyses and figures to evaluate the model-derived productivity changes in terms of mean, inter-annual variation and trend using these data.

Authors' Response: Thanks for this useful suggestion. We have evaluated the temporal changes in AGB at the six long-term field experimental sites and added a figure showing these temporal variations at site scales (Fig. 7). In general, the long-term field observations also show large inter-annual variabilities in the grassland biomass (Fig. 7) and can support our predicted temporal biomass dynamics at the regional scale (Fig. 6). For example, at four of the six sites, AGB showed a general decreasing trend (Fig. 7). We have included these results in the revised MS (Line 226-228).

Minor comments: Change the error of AGB unit "ka ha-1" in Abstract Line 17.

Authors' Response: Modified accordingly (Line 18).

A table showing the details of environmental drivers might be helpful.

Authors' Response: We have previously summarized the details of environmental drivers in the Supplement Table S1 in the last submission. In the revision, we have moved this table to the main text as Table 1 (the previous Table 1 has been accordingly updated to Table 2).

In the future projection, current grazing intensity was kept stable, while it would not be consistent with RCP simulations under RCP4.5 and RCP8.5. Some hypothetic scenarios are necessary.

Authors' Response: Thanks for pointing out this. We admit that this is one of the major uncertainty sources in the predicted AGB, we have included and discussed the associated uncertainties and limitations in the revised manuscript (Line 281-283).

References: Brookshire, E. N. J., and Weaver, T.: Long-term decline in grassland productivity driven by increasing dryness, Nature Communications, 6, 10.1038/ncomms8148, 2015. Brownlee, J.: Machine learning mastery with python, Machine Learning Mastery Pty Ltd, 100-120, 2016. Eldridge, D. J., and Delgadoâ ĂŘBaquerizo, M.: Continental‐scale impacts of livestock grazing on ecosystem supporting and regulating services, Land Degradation & Development, 28, 1473-1481, 2017. Griffiths, B. S., Spilles, A., and Bonkowski, M.: C:N:P stoichiometry and nutrient limitation of the soil microbial biomass in a grazed grassland site under experimental P limitation or excess, Ecological Processes, 1, 6, 10.1186/2192-1709-1-6, 2012. National Research Council: Grasslands and Grassland Sciences in Northern China, The National Academies Press, Washington, DC, 230 pp., 1992. Polley, H. W., Aspinwall, M. J., Collins, H. P., Gibson, A. E., Gill, R. A., Jackson, R. B., Jin, V. L., Khasanova, A. R., Reichmann, L. G., and Fay, P. A.: $CO_2$ enrichment and soil type additively regulate grassland productivity, New Phytologist, 222, 183-192, 10.1111/nph.15562, 2019. Wang, S., Zhang, Y., Ju, W., Chen, J. M., Ciais, P., Cescatti, A., Sardans, J., Janssens, I. A., Wu, M., Berry, J. A., Campbell, E., Fernández-Martínez, M., Alkama, R., Sitch, S., Friedlingstein, P., Smith, W. K., Yuan, W., He, W., Lombardozzi, D., Kautz, M., Zhu, D., Lienert, S., Kato, E., Poulter, B., Sanders, T. G. M., Krüger, I., Wang, R., Zeng, N., Tian, H., Vuichard, N., Jain, A. K., Wiltshire, A., Haverd, V., Goll, D. S., and Peñuelas, J.: Recent global decline of $CO_2$ fertilization effects on vegetation photosynthesis, Science, 370, 1295-1300, 10.1126/science.abb7772, 2020. Yang, Y., Fang, J., Pan, Y., and Ji, C.: Aboveground biomass in Tibetan grasslands, J Arid Environ, 73, 91-95, 2009. Zhang, Q., Buyantuev, A., Fang, X., Han, P., Li, A., Li, F. Y., Liang, C., Liu, Q., Ma, Q., Niu, J., Shang, C., Yan, Y., and Zhang, J.: Ecology and sustainability of the Inner Mongolian Grassland: Looking back and moving forward, Landscape Ecology, 35, 2413-2432, 10.1007/s10980-020-01083-9, 2020.

---

## Author Response (AR2)

Dear editor,

Thanks for your efforts in processing our MS. We have addressed the comments from the second reviewer as follows (the first reviewer didn't provide specific comments in this round). In addition, we have also thoroughly edited our language of the whole paper in this round.

We thank you again for your consideration of our paper and look forward to hearing from you soon.

Guocheng Wang et al.

**Referee #2**

I am happy that the authors considered the CO2 fertilization effect on the AGB. After considering this CO2 effect, there is a reverse in future changes of AGB. In the abstract (Line 24), the authors should emphasize AGB changes only from the perspective of climate change. For space-time substitution, I understood that it would be difficult to be resolved. But I would suggest adding more discussion and acknowledging the limitations.

**Authors' Response:** Thank you very much for your understanding of our study. We have emphasized that the simulated AGB changes in the future is only based on the perspective of climate change in the Abstract (Line 24). The possible uncertainties induced by space-time substitution has also been discussed in the revision (Line 303-307).